# Can Electronegative LDL Act as a Multienzymatic Complex?

**DOI:** 10.3390/ijms24087074

**Published:** 2023-04-11

**Authors:** Sonia Benitez, Núria Puig, José Rives, Arnau Solé, José Luis Sánchez-Quesada

**Affiliations:** 1Cardiovascular Biochemistry Group, Research Institute of the Hospital de la Santa Creu i Sant Pau (IIB Sant Pau), 08041 Barcelona, Spain; 2CIBER of Diabetes and Related Metabolic Diseases (CIBERDEM), Instituto de Salud Carlos III, 28029 Madrid, Spain; 3Biochemistry and Molecular Biology Department, Universitat Autònoma de Barcelona, 08193 Cerdanyola, Spain

**Keywords:** low-density lipoprotein, modified LDL, electronegative LDL, platelet-activating factor acetylhydrolase, phospholipase C, sphingomyelinase, ceramidase, LDL aggregation, inflammation

## Abstract

Electronegative LDL (LDL(−)) is a minor form of LDL present in blood for which proportions are increased in pathologies with increased cardiovascular risk. In vitro studies have shown that LDL(−) presents pro-atherogenic properties, including a high susceptibility to aggregation, the ability to induce inflammation and apoptosis, and increased binding to arterial proteoglycans; however, it also shows some anti-atherogenic properties, which suggest a role in controlling the atherosclerotic process. One of the distinctive features of LDL(−) is that it has enzymatic activities with the ability to degrade different lipids. For example, LDL(−) transports platelet-activating factor acetylhydrolase (PAF-AH), which degrades oxidized phospholipids. In addition, two other enzymatic activities are exhibited by LDL(−). The first is type C phospholipase activity, which degrades both lysophosphatidylcholine (LysoPLC-like activity) and sphingomyelin (SMase-like activity). The second is ceramidase activity (CDase-like). Based on the complementarity of the products and substrates of these different activities, this review speculates on the possibility that LDL(−) may act as a sort of multienzymatic complex in which these enzymatic activities exert a concerted action. We hypothesize that LysoPLC/SMase and CDase activities could be generated by conformational changes in apoB-100 and that both activities occur in proximity to PAF-AH, making it feasible to discern a coordinated action among them.

## 1. LDL Modification and Atherosclerosis

High levels of cholesterol in blood, specifically those associated with low-density lipoprotein (LDL), are the main cause of atherosclerotic cardiovascular disease, a pathological process triggered by the overwhelming accumulation of lipids within the arterial wall that leads to inflammation [1]. Indeed, LDL is a major causal agent in atherogenesis because these particles transport up to 80% of plasma cholesterol and are the source of most of the cholesterol accumulated in atherosclerotic lesions [2]. However, LDL presents few, if any, atherogenic characteristics in its native state: native LDL is neither inflammatory nor apoptotic, is not strongly retained by arterial proteoglycans, and does not induce, at low concentrations, the massive lipid accumulation in the cytoplasm of macrophages. The accumulation of cholesterol in macrophages only occurs at very high concentrations of native LDL by macropinocytosis [3]. Thus, there is a general consensus that LDL particles must suffer some kind of modification that confers on them their atherogenic properties [4]. The oxidative modification of both the lipid and protein moieties of LDL is, by far, the most studied modifying mechanism, although other biological processes, such as proteolysis, lipolysis, glycosylation, carbamylation, and desialylation, have also been implicated in LDL modification [5,6]. Another fact that has been generally accepted for years is that LDL is modified not during plasma circulation but when it crosses the endothelium and enters the arterial intima. However, growing evidence demonstrates that modified LDL particles, either oxidized or displaying other alterations, can be detected in plasma and increase under pathological conditions [7]. A common characteristic of the different modifications altering LDL is an increase in the negative net charge of the particles. Indeed, the presence of a heterogeneous group of modified LDL particles called electronegative LDL (LDL(−) or L5) has been reported in blood [8,9,10]. An important aspect to take into account is that several processes can lead to the formation of different subpopulations of LDL(−), and, consequently, their composition and physicochemical characteristics may differ. For example, the presence of light and larger LDL(−) particles has been described in patients with familial hypercholesterolemia, while in hypertriglyceridemia LDL(−) particles tend to be small and dense [11].

## 2. Association of LDL(−) with Disease

By taking advantage of the increased electronegativity of modified LDL, it is possible to isolate these particles from the bulk of LDL via methodologies based on the separation of molecules by means of their electric charge, such as capillary isotachophoresis (CIP) or anion-exchange chromatography (AEC) [12,13]. These methods have been used to quantify the relative proportion of LDL(−) in different groups of subjects with increased cardiovascular risk. Numerous studies have shown that the proportion of LDL(−) is increased in patients with familial hypercholesterolemia, hypertriglyceridemia, type 1 and 2 diabetes, obesity, metabolic syndrome, nonalcoholic fatty liver disease, autoimmune rheumatic disease, ischemic peripheral arterial disease, chronic kidney disease, and mental illness (reviewed in [9,10,14,15,16,17,18,19]). Moreover, LDL(−) increases during the acute phase of myocardial infarction or ischemic stroke (reviewed in [20]). Table 1 summarizes the pathological situations in which the proportion of LDL(−) increases.

These findings have been extensively reviewed by several authors and unequivocally highlight the relation between LDL(−) and atherothrombosis. Such a relationship is also suggested by several atherogenic properties ascribed to LDL(−) in in vitro studies, as will be shown in Section 3. However, there is no direct evidence as to whether LDL(−) is a mere biomarker of the presence of underlying atherosclerotic lesions or, instead, if it plays an active role in the origin and development of atherosclerosis [16]. Perhaps both phenomena occur simultaneously. Accordingly, different hypotheses have been proposed to explain the origin of plasma LDL(−). On one hand, the plasma LDL(−) burden may arise from plasma modifications of native LDL or from a defective catabolism of VLDL, situations that would occur more frequently in the presence of metabolic alterations [9]. On the other hand, it has also been speculated that the presence of abundant atherosclerotic plaques with trapped electronegative lipoproteins could contribute to the pool of LDL(−) in blood, especially when a plaque partially or totally ruptures [21,22,23]. In turn, these modified electronegative particles generated in plasma and/or released from established lesions could contribute to the development of new atherosclerotic plaques in a vicious circle, as suggested by their atherogenic properties, which are described below.

## 3. Atherogenic Properties of LDL(−)

AEC is currently the only preparative method for isolating LDL(−) from blood, although CIP and agarose electrophoresis have been used for analytical quantification [12,24,25]. In the pioneering studies of Pietro Avogaro and coworkers, who were the first to isolate LDL(−), several potentially pro-atherogenic properties were attributed to these modified lipoproteins [21,26]. The first studies mainly focused on their effects on endothelial cells and showed that LDL(−) had the capacity to induce the release of chemokines, such as IL8 and MCP1, from these cells [27,28,29]. Further studies have greatly expanded the number of inflammatory molecules for which expression is stimulated in endothelial cells via the activation of lectin-like oxidized low-density lipoprotein receptor-1 (LOX-1) [8,30,31,32,33,34,35,36,37,38]. Similarly, LDL(−) also activates an inflammatory response in leukocytes, including lymphocytes, monocytes, and differentiated macrophages; in this case, it is the main receptor involved in the complex CD14-TLR4, which primes and activates the NLRP3 inflammasome [32,33,34,35]. The intracellular signaling pathways activated by LDL(−) in monocytes have been elucidated and include CD14-TLR4 activation, the cascade of kinases (P38-MAPK, PI3/Akt), and transcription factors (NF-kB, AP1, CREB), eventually leading to inflammatory cytokine release [39,40]. These inflammatory actions are mediated mainly by bioactive lipids, including oxidized phospholipids (oxPC), lysophosphatidylcholine (LysoPC), non-esterified fatty acids (NEFA), ceramide (Cer), and sphingosine (Sph) [29,35,40,41,42,43]. Some of these lipids are generated by the enzymatic activities discussed in this review. In addition, LDL(−) also acts on macrophages via LOX-1, inducing M1 polarization [44] or triggering IL-1β production [45].

One of the first atherogenic effects of LDL(−) described was its cytotoxic effect on endothelial cells [46]. Further studies have described in detail the signaling pathways leading to mitochondrial dysfunction and apoptosis in different cell types [36,38,47], including, in addition to the arterial wall cells named above, cardiomyocytes [48]. Moreover, LDL(−) favors the accumulation of triglycerides in cardiomyocytes, an effect that impacts myocardium function [49]. A similar effect of triglyceride accumulation has been described in macrophages, which also promote cell differentiation [50]. The latter effect may be promoted via the LDL(−)-induced release of growth factors (GM-CSF, FGF2) by endothelial cells and leukocytes. Additionally, LDL(−) favors the release of matrix metalloproteinase 9 by monocytes, thus playing a role in the instability of the plaque as well [51].

Finally, LDL(−) is prothrombotic. Although this characteristic has not been demonstrated in LDL(−) taken from healthy subjects or patients with chronic disease, when LDL(−) (generally called L5) is isolated during the acute phase of myocardial infarction or ischemic stroke, it has the capacity to trigger platelet aggregation favoring thrombosis [22,23].

## 4. Some Potentially Protective Properties of LDL(−)

Although much rarer than the pro-atherogenic properties described above, some anti-atherogenic capacities of LDL(−) have also been reported. Ziouzenkova et al. [52] showed that the expression of vascular cell adhesion molecule-1, which is mediated by LDL(−) in endothelial cells, was reversed by peroxisome proliferator-activated alpha agonists generated by lipoprotein lipase, inhibiting the activation of the inflammatory factor NF-kB. On the other hand, LDL(−) induces the release of the anti-inflammatory cytokine IL10 by monocytes and lymphocytes, thereby limiting the excessive production of other inflammatory molecules [53]. Both effects could be considered regulatory actions that avert an excessive inflammatory response of cells against LDL(−).

Another feature that could count as anti-atherogenic is the presence of proteins that, at least theoretically, would play protective roles, such as platelet-activating factor acetylhydrolase ((PAF-AH), also known as lipoprotein-associated phospholipase A2 (Lp-PLA2)) or apolipoprotein J (apoJ, also known as clusterin). PAF-AH is an anti-inflammatory enzyme with the role of degrading the bioactive lipid PAF or other oxidized phosphatidylcholines, inactivating the high inflammatory potential of these phospholipids. Therefore, the fivefold increased content of PAF-AH activity in LDL(−) compared to native LDL could be considered an anti-atherogenic trait [54]. In the same context, LDL(−) contains an increased content of apoJ [55], an extracellular chaperone that prevents the aggregation of numerous proteins in plasma, including LDL. It is likeliest that this high apoJ content is a response to prevent the aggregation of LDL particles and the subsequent precipitation of such aggregates [56].

Such putative protective actions of LDL(−) could counteract, to some degree, the atherogenic ones, representing a regulatory mechanism to avoid an overwhelmed inflammatory response. Therefore, the final action of LDL(−) may result from the balance between its pro-atherogenic and anti-atherogenic properties, which could differ depending on the physiological context [16].

More studies are needed to better define whether the actions of LDL(−) are pro- or anti-atherogenic,. For example, by using different ratios of native LDL and LDL(−), similar to those performed on different populations at a high risk of cardiovascular disease. Therefore, LDL(−) should be used not only from healthy subjects but also from patients with a high proportion of LDL(−), such as individuals with familial hypercholesterolemia or diabetes. In this regard, it should be noted that the concentration of LDL(−) in the arterial intima should presumably be higher than in plasma, due to the increased affinity of LDL(−) for arterial proteoglycans.

## 5. Structural Alterations in LDL(−)

LDL particles are mainly composed of surface polar lipids (20–25% phospholipids and 10–15% free cholesterol), a lipid core with neutral lipids (where cholesterol esters are approximately 45–50% of the particle mass and triglycerides comprise 5–10%), and one molecule of a very large protein, apolipoprotein B-100 (apoB-100, approximately 25% of the mass), which is embedded in the surface but with hydrophobic regions in contact with the core [57]. Characteristic features of LDL(−) include the presence of structural abnormalities in apoB-100 and the disposition of lipids on the surface of the lipoprotein [58]. Indeed, circular dichroism studies have revealed a partial loss of a secondary structure in apoB [59,60,61], and immunochemical analysis [62] and 2D nuclear magnetic resonance [63] have shown that the spatial tertiary conformation of apoB-100 is also altered in LDL(−). Beyond alterations in apoB-100, another particularity of LDL is the presence of minor amounts of other apolipoproteins, including apoA-I, apoA-II, apoE, apoC-III, apoC-II, apoJ, apoF, and apoD, among others [55,64]. It must be taken into account that none of these proteins is present in all LDL(−) particles, and that the molar content versus apoB is always less than one, the most abundant being apoC-III, which would be present in one out of every three LDL(−) particles. The presence of different minor proteins is one of the main factors of LDL(−) heterogeneity and could play an important role in the pro- or anti-atherogenic properties of these particles. This aspect must be addressed in future studies since the role that each specific protein plays in LDL(−) is rather unknown. In addition, the possible coexistence of two or more apolipoproteins in the same LDL(−) particle must also be examined. Whether these proteins are related to the conformational alterations of apoB in LDL(−) has not been established. However, it is known that apolipoproteins, such as apoA-I, apoJ, and apoE, prevent the aggregation of LDL by phospholipases or intense agitation [56,65,66]. In fact, it cannot be ruled out that the formation of hydrophobic patches could be a mechanism for attracting apolipoproteins with amphipathic regions. Regarding lipids, the packaging of surface polar lipids in LDL(−) presents a disordered structure because of abnormal polarity [59,67]. This is probably related to an altered content of triglycerides in the core of the particles and of NEFA and Cer on the surface. This alteration in lipid packaging could facilitate the action of the phospholipolytic activities reported in LDL(−), which are discussed in the next paragraphs.

Table 2 summarizes the structural alterations in LDL(−). There are multiple consequences of these alterations that dramatically affect the biological characteristics and metabolic fates of these particles.

First, the poor affinity of LDL(−) for the receptor of native LDL (LDLr) comes from the altered ionization state of a population of lysine residues in apoB-100 involved in recognition by the LDLr [63]. This results in diminished clearance via the LDLr and could prolong the lifetime of LDL(−) in blood, favoring its modification by different mechanisms (oxidation, glycosylation, and desialylation); however, this possibility should be considered with caution, because, although our group described that LDL(−) has a lower affinity for the LDLr, and the increase in electronegative charge is not sufficient to be recognized by type A scavenger receptors in macrophages [68], we also know that LDL(−) binds to LOX1 or the CD14/TLR4 complex, which could facilitate its plasma clearance by these pathways.

Second, LDL(−) has a high tendency to aggregate [59,69], a phenomenon that has been implicated in the triggering phases of atherogenesis. This high susceptibility to aggregation can be attributed to two particularities of LDL(−). On one hand, it has been reported that the misfolding of apoB-100 acts as a priming factor that promotes the formation of globular aggregates that evolve to form fern-like structures [59,70]. On the other hand, the aggregation process can be triggered by the presence of hydrophobic spots on the surface of LDL(−) particles, which are probably related to the poor packaging of surface phospholipids [58]. In this context, LDL(−) presents an increased content of ceramide (Cer) [40], which could account for the loss of surface packaging that forms hydrophobic spots. The increased content of NEFA in LDL(−) is another factor contributing to its higher susceptibility to aggregation [41,68,71]. The presence of these features of both the protein and lipid moieties means that LDL(−) is not only more susceptible to aggregation but also that it has the capacity to induce the aggregation of native LDL particles via a mechanism that could be considered amyloidogenic [59].

A third effect of the altered conformation of apoB-100 in LDL(−) is an increased affinity for arterial proteoglycans compared to native LDL [61], which is considered the key factor in the subendothelial retention of lipoproteins. Indeed, our group demonstrated by different methods (colorimetric assay, high-speed precipitation, and affinity chromatography) that the binding affinity to commercial glycosaminoglycans and proteoglycans isolated from the arterial wall was fourfold increased in LDL(−) compared to native LDL. Accordingly, LDL(−) could act as a priming factor for atherogenesis, which, in addition to triggering the aggregation of native LDL, could facilitate its subsequent entrapment by proteoglycans of the arterial wall. Altogether, the higher retention in the arterial wall enables LDL(−) to elicit deleterious actions in that microenvironment, thereby contributing to the progression of atherosclerosis.

## 6. Enzymatic Activities Associated with LDL(−)

Besides apoB-100, LDL particles present minor quantities of other apolipoproteins and enzymes. This content is much higher in LDL(−) (up to 5% of the total protein mass) than in native LDL (below 1% of the protein mass) [55]. Among these minor proteins, as discussed above, is the enzyme PAF-AH, for which activity and mass is increased four- to sixfold in LDL(−) compared to native LDL [54]. Gaubatz et al. suggested that the increased content of PAF-AH in electronegative LDL is related to the decreased size of these particles, which would allow a conformational change in the carboxyl-terminal of apoB, enabling the binding of PAF-AH [72]. Notably, the activity of PAF-AH in small LDL particles is increased (10-fold K_m_ and 150-fold V_max_) compared to in large LDL particles [73]. Interestingly, small LDL particles are more likely to suffer oxidative modifications [74]. PAF-AH is an anti-inflammatory enzyme that has the primary role of inactivating the bioactive lipid PAF by hydrolyzing the methyl group at the sn-2 ether bond and attenuating its high biological activity [75]; however, in addition to this activity, PAF-AH also degrades the short-chain fatty acids in position sn-2 in the ester bond of phosphatidylcholine (PC), which are generated by the oxidation of polyunsaturated fatty acids. These oxidized phospholipids (oxPC) are formed during the early stages of LDL oxidation and are highly inflammatory; accordingly, their degradation by PAF-AH should reduce the inflammatory activity of oxidized LDL. However, there is controversy about the protective action of PAF-AH because, as a result of the degradation of oxPC, other lipids are formed, specifically lysoPC and short-chain oxidized NEFA (oxNEFA) [76,77]. Short-chain oxNEFA are oxidized fragmented acyl chains formed after the alkoxyl rearrangement of the unsaturated fatty acids at the sn2 position in the glycerol backbone of the phospholipid. There are multiple oxNEFAs, depending on which polyunsaturated fatty acid has undergone an oxidative attack (linoleic and arachidonic acids are the most abundant in LDL) and at what position of the aliphatic chain it has been cleaved. Some studies report that both lysoPC and oxNEFA also display inflammatory capacities if they remain retained in the LDL particle, although they are not as inflammatory as oxPC [78]. Indeed, part of the LDL(−)-induced inflammatory effect on endothelial cells and leukocytes has been attributed to an increased content of both lipid species [41,79].

A second enzymatic activity associated with LDL(−) is phospholipase C-like (PLC-like) activity [69,80]. This activity, which is absent in native LDL, can degrade the polar head of choline-containing phospholipids and is especially effective in the hydrolysis of lysoPC and, to a lesser extent, sphingomyelin (SM), with its activity on PC being almost residual. The effects of the PLC-like activity present in LDL(−) are very different depending on the degraded substrate; thus, when PLC-like activity hydrolyzes LysoPC, the effect could be anti-inflammatory, since the resultant products, monoacylglycerol and phosphorylcholine, are not known to display inflammatory potential. In contrast, the degradation of SM by PLC-like activity (in this case, it would be SMase-like activity) that yields Cer implies a clear pro-atherogenic action since it triggers the aggregation not only of LDL(−) itself but also of native LDL particles [9]. Cer molecules remain on the surface of the lipoprotein and form Cer-enriched spots with a hydrophobic character. The presence of these hydrophobic spots on the lipoprotein surfaces of different LDL particles favors their interaction and triggers their aggregation. Interestingly, the presence of apoJ, an apolipoprotein with chaperone activity, prevents the aggregation of LDL(−) [81]. Another consequence of the formation of Cer is a contribution to the inflammatory and apoptotic effects of LDL(−) on monocytes [40].

The third enzymatic activity associated with LDL(−) is ceramidase-like (CDase-like) activity, which is described in this same Special Issue [82]. The CDase-like activity is absent in native LDL. LDL(−) can degrade Cer contained in both LDL(−) particles and external substrates, yielding sphingosine (Sph) and NEFA. The physiological effects of this CDase-like activity in LDL(−) have not been defined but, apparently, could limit the pro-aggregating and pro-apoptotic effect of the PLC-like activity by regulating the content of Cer, a molecule strongly involved in lipoprotein aggregation and apoptosis [83,84]. On the other hand, the Sph yielded has been described as strongly contributing to the inflammatory properties of LDL(−) [85]. Therefore, further studies are needed to determine the effect that this CDase activity could have on the pro-atherogenic or anti-atherogenic characteristics of LDL(−). Taken together, the biological effect of LDL(−) depends on the relative contribution of each enzymatic activity, which could vary as a consequence of several factors, such as lipid composition and apoB conformation.

Table 3 summarizes the substrates and products of the different enzymatic activities associated with LDL(−).

## 7. Origin of PLC-Like and CDase-Like Activities Associated with LDL(−)

In contrast to PAF-AH, the origin of the PLC-like and CDase-like activities of LDL(−) is still unknown. Proteomic analyses have found no protein or apolipoprotein with enzymatic activity able to degrade the polar head group of lysoPC or SM or the fatty acid ester bond of ceramide [55]. Although it is possible that the amount of such hypothetical enzymes is so low that current proteomic methodologies cannot detect them, the likeliest possibility is that apoB-100 itself is responsible for this activity [58]. Since the apoB-100 sequence is the same in LDL(−) and native LDL, this possibility implies the existence of some kind of conformational difference in LDL(−). ApoB-100 is a very large protein (550 kDa) with a highly dynamic structure that allows it to adapt to the alterations undergone by the lipoprotein particle, from its synthesis in the liver and release into the bloodstream in the form of VLDL particles (>80 nm in diameter) to its plasmatic maturation into LDL particles (of 20–25 nm). As noted above, conformational changes are part of the nature of apoB-100 and are mainly driven by changes in the lipid content of LDL.

Holopainen and coworkers, who reported SMase activity (similar to PLC-like activity) in total LDL for the first time, suggested that a catalytically active His–Ser–Asp triad, common to a wide range of lipolytic enzymes, is also present in apoB-100, and located this triad between residues 2155 and 2359 in the α2 region of apoB-100 [84,86]. Interestingly, Parasassi et al. predicted an aggregation-prone region in the same position as apoB-100 (residues 2075 to 2575) [59] and suggested that this epitope should be exposed in LDL(−), although they did not relate this region to any enzymatic activity. More recently, C.H. Chen’s group made a sequence alignment and protein structure prediction in comparison to *Bacillus cereus* SMase, concluding that residues participating in the SMase-like activity could be in the 2021–2377 region [87]. In addition, these authors found that serine residues S^1732^, S^1959^, S^2378^, S^2408^, and S^2429^ are highly glycosylated in LDL(−), which may be related to the presence of SMase activity [87]. The findings of these investigators point to putative conformational changes in the α2 domain of apoB-100 to differentiate between native LDL and LDL(−). However, these studies are based on sequence predictions, and no experimental data support the different conformations of the α2 domain in apoB-100 between native LDL and LDL(−).

In contrast, using an experimental immunochemical approach, our group reported that the main conformational differences of apoB-100 between native LDL and LDL(−) involve both the N-terminal and C-terminal extremes of the apoB-100 sequence [62]. We used 28 monoclonal antibodies to recognize different epitopes throughout the apoB-100 sequence and analyze the conformation of apoB-100 in LDL(−). This study revealed that both extremes of apoB-100 in LDL(−) had higher immunoreactivity to specific monoclonal antibodies (Bsol10 and Bsol14 for the N-terminal, and Bsol2 and Bsol7 for the C-terminal) than in native LDL. In contrast, the antibodies recognizing the α2 domain of apoB-100 showed no alterations in immunoreactivity in this region between native LDL and LDL(−). The abnormal conformation of the N-terminal resembles the immunoreactivity of native LDL lipolyzed by bacterial SMase (binding to the antibodies Bsol10 and Bsol14). Both LDL(−) and SMase-modified LDL present increased binding to arterial proteoglycans, suggesting that this alteration in the N-terminal region could be involved in the retention of these lipoproteins in the subendothelial space. On the other hand, it is known that Bsol2 and Bsol7 recognize with increased affinity the C-terminal extreme of apoB-100 in oxidized LDL particles [88], which suggests the involvement of oxidative processes in the formation of LDL(−).

Assuming that a conformational change in the structure of apoB-100 is necessary for PLC-like and CDase-like activities to emerge in LDL(−), and based on the results of the immunochemical study, it is tempting to speculate that these enzymatic activities could be located in one or both of the terminal extremes of apoB-100. This hypothesis is expanded upon below. However, the demonstration of the implication of both terminal extremes of apoB-100 in the enzymatic activities ascribed to LDL(−) requires further physicochemical studies.

## 8. Could the Enzymatic Activities Associated with LDL(−) Act Cooperatively as a sort of Enzymatic Complex?

Whatever the origin of PLC-like and CDase-like activities, the coincidence of the substrates and products of these activities and those of PAF-AH paints a picture in which a complementary effect is suggested. Figure 1 depicts a scheme of the putative cooperative actions of these enzymatic activities, which can be divided into three stages.

In the first stage, upon an oxidative injury generating highly inflammatory oxPC, PAF-AH could constitute a first-line defense by degrading oxPC to form LysoPC and short-chain oxNEFA. PAF-AH binds with greater affinity to dense LDL particles, which are more electronegative and highly susceptible to oxidation [72]. However, the binding of PAF-AH would not be sufficient to hamper the deleterious effects of oxidation since, although LysoPC and short-chain oxNEFA have lower inflammatory capacity than oxPC, they are still inflammatory (reviewed in [89]).

Oxidation is known to alter the conformation of apoB-100; for instance, it increases the reactivity of Bsol2 and Bsol7 antibodies that recognize the C-terminus of apoB-100 [88]. Here would begin the second stage, in which the oxidative injury could promote a change in the conformation of both the N-terminal and C-terminal extremes of apoB-100, boosting the emergence of PLC-like activity. This activity would degrade LysoPC, decreasing the inflammatory capacity of LDL(−). The flip part of PLC-like activity is that it also has the capacity to hydrolyze SM. The hydrolysis of SM yields Cer, which remains bound to the LDL particle due to its hydrophobic nature, and NEFA, which presumably leaves the LDL particle in the presence of albumin. Subsequently, Cer-rich hydrophobic spots are formed on the surface of the lipoprotein, favoring the aggregation of LDL. In addition, Cer could also contribute to LDL(−)-induced inflammatory and apoptotic effects. Altogether, these effects derived from SMase-like activity would be potentially hazardous.

It is known that SM hydrolysis alters the conformation of apoB-100 (for instance, increasing the reactivity of the N-terminal domain with Bsol10 and Bsol14) [62], which could induce the appearance of CDase-like activity in LDL(−). This conformational change would initiate the third stage. CDase-like activity would degrade Cer to form Sph and NEFA [82], which would decrease the formation of Cer-rich hydrophobic spots, thereby preventing related actions, including increased susceptibility to aggregation, inflammation, and apoptosis. Whether the emergence of PLC-like and CDase-like activities requires the interaction between both terminal extremes of apoB-100 or they appear in one or another extreme is unclear and requires further investigation.

An important question to consider is whether the interaction between the three phospholipolytic activities associated with LDL(−) and their substrates could feasibly occur as a chain reaction; however, for this to happen, the different enzymatic activities would have to be close to allow an easy transfer of the molecules produced by some enzymatic activities, which will be substrates for other activities. Several studies in the literature support this possibility. A study by Stafforini et al., in which lipoproteins containing truncated forms of apoB were expressed, demonstrated that the carboxyl terminus of apoB plays a key role in the association of PAF-AH with LDL [90]. In accordance with this, Gaubatz et al. proposed that PAF-AH binds to the C-terminal α-helical domains in the apoB-100 of small dense LDL(−), which is desorbed into the aqueous phase [72]. On the other hand, immunoelectron cryomicroscopy has shown that the two N- and C-terminal ends of apoB-100 are in close proximity in LDL [91]. As discussed above, N-terminals, C-terminals, or both extremes could be involved in the PLC-like activity ascribed to LDL(−). Therefore, LysoPC generated by PAF-AH activity may be easily accessible, allowing it to be degraded by PLC-like activity. This would facilitate the concerted action of the PLC-like and CDase-like activities, enabling the degradation of Cer yielded by SMase activity. This hypothesis suggests that the enzymes contained in LDL(−) may act in an orchestrated manner to counteract an excessive inflammatory effect and modulate the formation of aggregates.

Another relevant issue that must be considered is identifying in which subset of LDL(−) particles the proposed multienzyme complex could form. Since LDL(−) is highly heterogeneous, it is most likely that this phenomenon only could occur in a few LDL(−) particles. According to Gaubatz et al., the enzyme PAF-AH is only present in the 1% of the densest LDL particles [72], which presumably contribute the most to the bulk of LDL(−). Consequently, only a small part of the LDL(−) could form the multienzyme complex hypothesized here. This is an aspect that needs to be explored in future studies. In the same way, it will be necessary to analyze whether the presence of other minor proteins has some kind of role in the emergence of the enzymatic activities that our hypothesis proposes.

It is also relevant that the proposed hypothesis invokes a reciprocal interaction between the protein and lipid moieties of the lipoprotein to explain the amyloidogenic behavior of LDL(−). Thus, alterations in surface lipids, either by oxidation or other modifications, would cause the emergence of enzymatic activities, which, in turn, would alter the lipid composition. Overall, these alterations would determine the functionality of LDL(−) and its pathophysiological behavior.

The demonstration of this hypothesis requires the future development of complex experiments. The use of monoclonal antibodies specific to each terminal end of apoB can be very useful. On the one hand, the proximity of the two ends of apoB and PAF-AH in LDL(−) should be demonstrated by immunoelectron cryomicroscopy experiments. On the other hand, an attempt could be made to inhibit the enzymatic activities described throughout this review. However, the fact that the specificity of the monoclonal antibodies used so far for LDL(−) is not complete (although, with lower affinity, they also bind to native LDL) could lead to misleading results. A possible alternative to monoclonal antibodies would be the development of specific aptamers. Aptamers are fragments of RNA or DNA that recognize specific epitopes of proteins or other molecules with great specificity [92]. Obtaining one or several aptamers capable of inhibiting enzymatic activities would be very useful to define the specific role of these activities. Alternatively, the search for low-molecular weight phospholipase inhibitors or other hydrophobic molecules (such as 17-β-estradiol or related hormones [60]) capable of inhibiting these enzymatic activities would be of great help to confirm the proposed hypothesis.

## 9. Conclusions

The hypothesis presented here suggests that the different lipolytic enzymatic activities associated with LDL(−) could act in a concerted manner as a sort of multienzymatic complex. Of course, this “complex” would not exactly fit to the conventional definition of multienzyme complexes, in which physical contact stemming from the protein–protein interaction between the enzymes involved in the pathway is necessary. This hypothesis is based on the complementarity between the different substrates and products generated by each enzymatic activity as well as the feasibility that these activities occur within a certain vicinity in the LDL particle.

It is difficult to interpret the effect that the consecutive actions of these enzymatic activities may have on the function of LDL, although the evidence points to a mechanism for avoiding the excessive modification of LDL and the ensuing atherogenic effects. In particular, the coordinated action of these enzymatic activities could prevent LDL(−) from becoming excessively inflammatory and large lipoprotein aggregates from forming. However, a clear account of the origin of and mechanisms involved in these enzymatic activities would require further investigation. Another question that arises from this scheme is whether LDL(−) is pro-atherogenic or a physiological response to limit injury in the arterial wall. These open questions warrant further studies to define the role of LDL(−) in atherosclerotic disease.

## Figures and Tables

**Figure 1 ijms-24-07074-f001:**
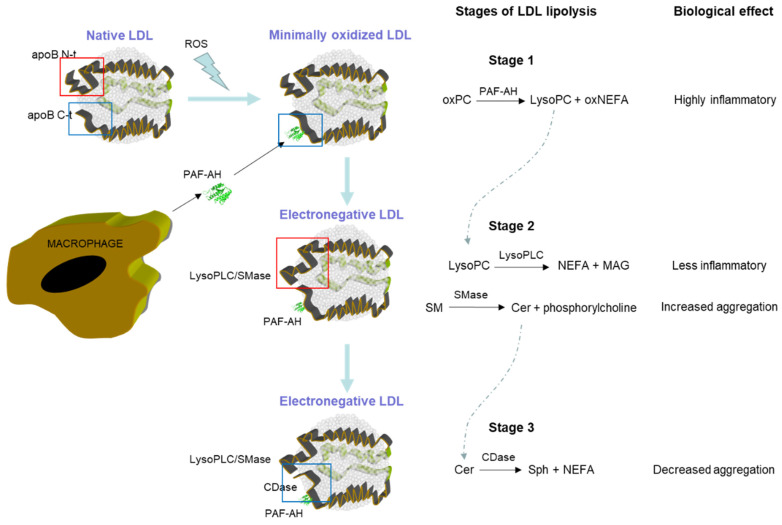
Scheme of the putative cooperative actions of PAF-AH, LysoPLC, and CDase activities. Stage 1: The attack of reactive oxygen species (ROS) on phosphatidylcholine favors the appearance of oxidized phospholipids (oxPC) that would act as a substrate for PAF-AH coming from macrophages, yielding LPC and oxNEFA. Stage 2: Minimal oxidation of LDL would promote a conformational change in the N-terminus of apoB (red square) causing the appearance of LysoPLC/SMase-like activity. The action of this enzymatic activity on the SM would produce the formation of Cer. Stage 3: A second conformational change at the C-terminus of apoB (blue square) could favor the appearance of CDase-like activity, which would degrade Cer, forming Sph and NEFA as the final products. Notably, this is just one of the possible cooperation schemes between the different enzymatic activities. With the current experimental information, we do not know if the SMase and CDase activities appear at the C-terminus or the N-terminus, or even if the interaction of both ends is necessary for these activities to emerge. Similarly, the proposed sequence of stages could be different or occur simultaneously.

**Table 1 ijms-24-07074-t001:** Situations with increased LDL(−) proportion.

Nonpathological	Chronic Diseases	Acute Phase Diseases
- Postprandial phase - Heavy aerobic exercise	- Chronic smoking - Type 1 diabetes - Type 2 diabetes - Familial hypercholesterolemia - Hypertriglyceridemia - Obesity - Metabolic syndrome - Ischemic peripheral arterial disease - Autoimmune rheumatic disease - Chronic kidney disease - Nonalcoholic fatty liver disease	- Myocardial infarction - Ischemic stroke

**Table 2 ijms-24-07074-t002:** Summary of structural alterations in LDL(−).

Component	Alteration
Proteins	- Partial loss of secondary structure in apoB-100. - Altered spatial tertiary conformation in apoB-100. - Decreased number of active surface-accessible lysine residues in apoB-100. - Highly *O*-glycosylated serines in apoB-100. - Increased content of other minor apolipoproteins.
Lipids	- A disordered structure of surface packaging. - Increased triglyceride, Cer, Sph, and NEFA content.

**Table 3 ijms-24-07074-t003:** Substrates and products of enzymatic activities in LDL(−).

Enzymatic Activity	Substrate	Product
PAF-AH activity	Oxidized PC	LysoPC Oxidized short-chain NEFA
PLC-like activity	LysoPC	Monoacylglycerol Phosphorylcholine
SMase-like activity	SM	Ceramide Phosphorylcholine
CDase-like activity	Ceramide	Sphingosine NEFA

## Data Availability

Not applicable.

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
