# Peer review of "Can Electronegative LDL Act as a Multienzymatic Complex?"

_ijms, 2023, doi:10.3390/ijms24087074_

Round 1

Reviewer 1 Report (Previous Reviewer 2)

the authors have satisfactorily responded to my comments

Author Response

Thank you for your kind comments

Reviewer 2 Report (Previous Reviewer 3)

I have no additional comments.

Author Response

Thank you for your kind comments

This manuscript is a resubmission of an earlier submission. The following is a list of the peer review reports and author responses from that submission.

Round 1

Reviewer 1 Report

In their review article entitled “Can electronegative LDL act as a multienzymatic complex?”, Benitez et al have put forward a thought-provoking proposal of considering electronegative LDL (LDL(-)) as ‘multienzyme complexes’. LDL(-) is a minor form of LDL present in blood that is present in increased amounts in pathologies with increased cardiovascular risk. The enzymes involved are platelet activating factor acetylhydrolase (PAF-AH) which degrades oxidized phospholipids, type C phospholipase-like (lysoPLC) activity that acts on lysophospholipids, sphingomyelinase-like activity (SMase) that acts on sphingomyelins and ceramidase (CDase) activity that acts on ceramides. The authors propose that together these enzymes exert a concerted action since they are all located on one large lipoprotein particle.

This reviewer finds the proposal both provocative and stimulating since they have presented a scenario with LDL(-) wherein the products generated by one enzyme serve as substrates for the another enzyme in the vicinity. However, insofar as the conventional definition of multienzyme complexes go, a physical contact involving protein-protein interaction between the enzymes involved in the pathway may be warranted. Such a proximity would allow direct channeling of products from the active site of one enzyme to serve as substrate in the active site of the next enzyme, thereby increasing the efficiency and speed of the overall process. There is insufficient evidence presented in the review for such a physical contact between these enzymes at this point. The LDL(-) is a massive particle (25-50 nm diameter) composed of >50 different proteins and hundreds of lipid species.

That said, the LDL(-) particle presents a confined/limited environment that can house the components of a multienzyme complex. The predominantly hydrophobic environment presented in the particle interior lends itself well for diffusion of intermediates between active sites of participating enzymes. This reviewer recommends that the authors include caveats such as these and include quantitative kinetic data of the enzymes involved to support the concept of LDL(-) being a multienzyme complex. Such analysis of reported data may help in obtaining support for the proposed concept of LDL(-) possiblly being a multienzyme complex.

PLC itself can cause LDL aggregation. The authors should describe and specify their views on how the presence of minor amounts of apolipoproteins like apoAI, apoAII, apoE, apoCIII etc contribute to structural alterations in LDL(-) (Page 4, Section 5). The authors could include citations regarding the in vitro effect of PLC on LDL; for example, it has been demonstrated that PLC addition causes LDL aggregation (FEBS Lett (1993) 316, 27), which can be prevented by small exchangeable apolipoproteins including apoE (BBA Proteins and Proteomics (2018) 1866, 165). These support the authors notion that the hydrophobic patches generated by the PLC-like activity leads to LDL (or LDL(-)) aggregation and that it may be responsible for recruiting proteins like apoE that are not normally found on LDL (Page 6, lines 9-12).

The statement “The physiological effects of this CDase-like activity in LDL(-) have not been defined but, apparently, could limit the pro-aggregating and pro-apoptotic effect of the PLC-like activity by regulating the content of Cer” should be supported by citations.

The statement “Regarding lipids, the packaging of surface polar lipids in LDL(-) presents a disordered structure because of abnormal polarity” is vague. Is it possible that abnormal polarity leads to structural defects in the lipid monolayer, which in turn leads to increased access of lipases. The increased lipase action could lead to altered (decreased?) triglycerides content in the core. The authors should re-phrase and/or expand this part.

The statement “….. the poor affinity of LDL(-) for the receptor of native LDL (LDLr) comes from the altered ionization state of a population of lysine residues in apoB-100 involved in recognition by the LDLr [56]. This results in diminished clearance and a prolonged lifetime of LDL(-) in blood and favors its modification by different mechanisms (oxidation, glycosylation, and desialylation)” is not necessarily true. Upon modification, the particles now become ‘eligible’ to bind to and be cleared by the large family of scavenger receptors. This part should be addressed.

Author Response

REVIEWER 1

In their review article entitled “Can electronegative LDL act as a multienzymatic complex?”, Benitez et al have put forward a thought-provoking proposal of considering electronegative LDL (LDL(-)) as ‘multienzyme complexes’. LDL(-) is a minor form of LDL present in blood that is present in increased amounts in pathologies with increased cardiovascular risk. The enzymes involved are platelet activating factor acetylhydrolase (PAF-AH) which degrades oxidized phospholipids, type C phospholipase-like (lysoPLC) activity that acts on lysophospholipids, sphingomyelinase-like activity (SMase) that acts on sphingomyelins and ceramidase (CDase) activity that acts on ceramides. The authors propose that together these enzymes exert a concerted action since they are all located on one large lipoprotein particle.

This reviewer finds the proposal both provocative and stimulating since they have presented a scenario with LDL(-) wherein the products generated by one enzyme serve as substrates for the another enzyme in the vicinity. However, insofar as the conventional definition of multienzyme complexes go, a physical contact involving protein-protein interaction between the enzymes involved in the pathway may be warranted. Such a proximity would allow direct channeling of products from the active site of one enzyme to serve as substrate in the active site of the next enzyme, thereby increasing the efficiency and speed of the overall process. There is insufficient evidence presented in the review for such a physical contact between these enzymes at this point. The LDL(-) is a massive particle (25-50 nm diameter) composed of >50 different proteins and hundreds of lipid species.

We thank the reviewer comments and we are glad that he/she considers our hypothesis provocative and stimulating. We agree with the reviewer that LDL(-) does not fully fit the conventional concept of a multienzyme complex. In this sense, we have moderated the idea of the multienzyme complex, making it clear that we are not talking about a multienzyme complex as such (page 10, lines 10-12). However, since the enzymatic activities described in the review coexist in the LDL(-) particle, it is plausible for the substrates of some activities to feed the action of the other enzymes. Further studies are needed to demonstrate that the hypothesized domains of apoB that might be responsible for enzyme activities may be close enough to enable the products of one activity to act as substrate for another. Probably, the main point is to demonstrate unequivocally that this neighborhood of inter-apoB domains and of PAF-AH occurs in LDL(-) but not in native LDL.

That said, the LDL(-) particle presents a confined/limited environment that can house the components of a multienzyme complex. The predominantly hydrophobic environment presented in the particle interior lends itself well for diffusion of intermediates between active sites of participating enzymes. This reviewer recommends that the authors include caveats such as these and include quantitative kinetic data of the enzymes involved to support the concept of LDL(-) being a multienzyme complex. Such analysis of reported data may help in obtaining support for the proposed concept of LDL(-) possibly being a multienzyme complex.

We appreciate the reviewer's comment. Indeed, the hydrophobic environment facilitates the diffusion of hydrophobic molecules between the different enzymatic activities, which could favor their coordinated action. It must also be considered that the ease with which lipids move on the surface of LDL means that such a close proximity between the epitopes of apoB and PAF-AH may not be so necessary. This point has been commented in (page 9, lines 1-4)

Regarding quantitative kinetic data, this is an aspect that will have to be addressed in future studies, although it implies important challenges, mainly due to the influence of the environment. The only enzyme involved in the hypothetical multienzyme complex that has been extensively studied from the kinetic point of view is PAF-AH. And despite being a very well-characterized enzyme, it has been observed that, depending on the lipoprotein fraction in which it is found, the Km can vary 10-fold and the Vmax 150-fold among the different lipoprotein subclasses (Tselepis et al. ATVB 1995;15:1764–1773). Thus, the kinetic study of coordinated action implies a great complexity derived from the different environments that lipoproteins can adopt. In this context, for a good characterization, it would be very helpful to have specific inhibitors of each enzymatic activity; unfortunately, we only have specific inhibitors for PAF-AH, but not for PLC-like or CDase-like activities.

PLC itself can cause LDL aggregation. The authors should describe and specify their views on how the presence of minor amounts of apolipoproteins like apoAI, apoAII, apoE, apoCIII etc contribute to structural alterations in LDL(-) (Page 4, Section 5). The authors could include citations regarding the in vitro effect of PLC on LDL; for example, it has been demonstrated that PLC addition causes LDL aggregation (FEBS Lett (1993) 316, 27), which can be prevented by small exchangeable apolipoproteins including apoE (BBA Proteins and Proteomics (2018) 1866, 165). These support the authors notion that the hydrophobic patches generated by the PLC-like activity leads to LDL (or LDL(-)) aggregation and that it may be responsible for recruiting proteins like apoE that are not normally found on LDL (Page 6, lines 9-12).

We thank the reviewer comments, this is a very important point. Some sentences have been included in this section discussing the possible role of apos on structural alterations of LDL(-) and in preventing the aggregation of LDL (page 4, lines 32-35). The reviewer's suggestion that the formation of hydrophobic patches could be a mechanism for attracting apolipoproteins with amphipathic regions seems very interesting to us, and has been included in the manuscript (page 4, lines 35-37), together with some new references.

The statement “The physiological effects of this CDase-like activity in LDL(-) have not been defined but, apparently, could limit the pro-aggregating and pro-apoptotic effect of the PLC-like activity by regulating the content of Cer” should be supported by citations.

Two references describing the ability of ceramide-rich lipid patches to promote lipoprotein aggregation and the role of ceramide in apoptosis have been included, and the sentence slightly modified (page 6, lines 37).

The statement “Regarding lipids, the packaging of surface polar lipids in LDL(-) presents a disordered structure because of abnormal polarity” is vague. Is it possible that abnormal polarity leads to structural defects in the lipid monolayer, which in turn leads to increased access of lipases. The increased lipase action could lead to altered (decreased?) triglycerides content in the core. The authors should re-phrase and/or expand this part.

The reviewer is right in pointing to a possible dual effect of the packaging of polar lipids, which could favor the access of lipases. However, experimentally what has been reported for LDL(-) is an increased content of Tg. Hence, it is improbable that a loss of packaging in LDL(-) results in increased triglyceride lipase accessibility. However, a greater accessibility of other lipases, including the intrinsic phospholipases displayed by LDL(-) is possible. This aspect has been included in the revised manuscript (page 4, lines 42-43).

The statement “….. the poor affinity of LDL(-) for the receptor of native LDL (LDLr) comes from the altered ionization state of a population of lysine residues in apoB-100 involved in recognition by the LDLr [56]. This results in diminished clearance and a prolonged lifetime of LDL(-) in blood and favors its modification by different mechanisms (oxidation, glycosylation, and desialylation)” is not necessarily true. Upon modification, the particles now become ‘eligible’ to bind to and be cleared by the large family of scavenger receptors. This part should be addressed.

We agree with the reviewer that the loss of affinity for the LDLr is usually accompanied by an increase in the affinity for scavenger receptors. But this is not always the case; our group described that LDL(-) has a lower affinity for LDLr, but the increase in electronegative charge is not sufficient to be recognized by scavenger receptors in macrophages of the P388 line (mainly SRA). In unpublished studies by our group, we have also seen that the binding of LDL(-) to CD36 is low. However, we do know that it binds to either LOX1 or CD14/TLR4. Studies from our group have reported the increased uptake of LDL(-) through CD14/TLR4 pathway and its involvement in mediating the induction of potentially atherogenic effects (refs 30 and 49 in the manuscript). In any case, this section has been rewritten to fit the reviewer's comment (page 5, lines 14-19).

Reviewer 2 Report

The present review summarizes and discusses several general aspects concerning electronegative LDL. Much information is already reported in other recent reviews, including at least one published by the same authors in 2019.

The original approach of the present review concerns the focus on the enzymatic activities associated with these lipoproteins and the discussion in paragraph 8 of the hypothesis summarized in the title, which however remains an object of speculation as underlined by the authors themselves.

Considering the recent reviews already present in the literature, to increase the reader's interest in a new review on this topic a careful updating of the bibliography is recommended

For example, as regards paragraph 2 on the associations with pathologies:

Vural H et al Clin Chim Acta. 2021

Akyol O et al Med Hypotheses. 2020

Chen CH J Diabetes Investig. 2020

Or for Paragraph 3 on atherogenic properties

Chang SF Inflammation. 2020

Chang PY J Nutr Biochem. 2021

The scheme in figure 1 is too simplistic and should be deeply  revised to better illustrate the hypothesis proposed by the authors

Author Response

REVIEWER 2

The present review summarizes and discusses several general aspects concerning electronegative LDL. Much information is already reported in other recent reviews, including at least one published by the same authors in 2019.

The original approach of the present review concerns the focus on the enzymatic activities associated with these lipoproteins and the discussion in paragraph 8 of the hypothesis summarized in the title, which however remains an object of speculation as underlined by the authors themselves.

Considering the recent reviews already present in the literature, to increase the reader's interest in a new review on this topic a careful updating of the bibliography is recommended

We thank the reviewer’ comment. According to his/her suggestion, the bibliography has been updated (9 new references), including the suggested references.

For example, as regards paragraph 2 on the associations with pathologies:

Vural H et al Clin Chim Acta. 2021

Akyol O et al Med Hypotheses. 2020

Chen CH J Diabetes Investig. 2020

Or for Paragraph 3 on atherogenic properties

Chang SF Inflammation. 2020

Chang PY J Nutr Biochem. 2021

The scheme in figure 1 is too simplistic and should be deeply revised to better illustrate the hypothesis proposed by the authors.

We agree with the reviewer that figure 1 is simple. This is because we have no structural data in the literature regarding the possible conformations that the terminal extremes of apoB could adapt in LDL(-), and then we have limited ourselves to making a very schematic drawing. However, we have included some changes in figure 1, and the figure legend has been modified accordingly. We hope this version will seem more suitable to the reviewer

Reviewer 3 Report

The authors summarized in this review the characteristics of electronegative LDL or LDL(-). LDL(-) is thought to be pro-atherogenic subfraction of LDL although the mechanisms for atherogenicity is not fully understood. The authors focused on this review the enzymatic activities present in LDL(-) and structural changes in apoB in LDL(-), and provided a new viewpoint to explain the characteristics of LDL(-). This review article is concise and well written, but I would like to mention 2 points that would improve this review.

Specific comments to the authors:

1.       Since this review focuses on possible importance of the 4 enzyme activities in LDL(-), the information that each enzyme (protein and or activity) in LDL(-) is higher than normal LDL are desirable.

2.       It will be very nice if more references from latest publications (recent ~5 years) are cited in this manuscript.

Author Response

REVIEWER 3

The authors summarized in this review the characteristics of electronegative LDL or LDL(-). LDL(-) is thought to be pro-atherogenic subfraction of LDL although the mechanisms for atherogenicity is not fully understood. The authors focused on this review the enzymatic activities present in LDL(-) and structural changes in apoB in LDL(-), and provided a new viewpoint to explain the characteristics of LDL(-). This review article is concise and well written, but I would like to mention 2 points that would improve this review.

We thank the reviewer for his/her kind comments.

Specific comments to the authors:

  1. Since this review focuses on possible importance of the 4 enzyme activities in LDL(-), the information that each enzyme (protein and or activity) in LDL(-) is higher than normal LDL are desirable.

We thank the reviewer comment. In the modified version of the manuscript, the differences in the enzymatic activities between LDL(-) and native LDL have been highlighted in chapter 6.

  1. It will be very nice if more references from latest publications (recent ~5 years) are cited in this manuscript.

According to the reviewer suggestion the bibliography has been updated (9 new references).

Round 2

Reviewer 1 Report

Satisfactory revisions

Accept the revised version

Reviewer 2 Report

 the revised version meets my demands